# Impact of Cigarette Smoking on the Risk of Osteoporosis in Inflammatory Bowel Diseases

**DOI:** 10.3390/jcm10071515

**Published:** 2021-04-05

**Authors:** Alicja Ewa Ratajczak, Aleksandra Szymczak-Tomczak, Anna Maria Rychter, Agnieszka Zawada, Agnieszka Dobrowolska, Iwona Krela-Kaźmierczak

**Affiliations:** Department of Gastroenterology, Dietetics and Internal Diseases, Poznań University of Medical Sciences, 61-701 Poznań, Poland; aleksandra.szymczak@o2.pl (A.S.-T.); a.m.rychter@gmail.com (A.M.R.); aga.zawada@gmail.com (A.Z.); agdob@ump.edu.pl (A.D.)

**Keywords:** Crohn’s disease, inflammatory bowel disease, ulcerative colitis, bone disorders, IBD clinical course, smoking, osteoporosis, bone mineral density

## Abstract

Cigarette smoking constitutes one of the most important modifiable factors of osteoporosis, as well as contributes to an early death, tumors, and numerous chronic diseases. The group with an increased risk of a lower bone mineral density are patients suffering from inflammatory bowel diseases. In fact, tobacco smoke, which contains more than 7000 chemical compounds, affects bone mineral density (BMD) both directly and indirectly, as it has an impact on the RANK-RANKL-OPG pathway, intestinal microbiota composition, and calcium–phosphate balance. Constant cigarette use interferes with the production of protective mucus and inhibits the repair processes in the intestinal mucus. Nicotine as well as the other compounds of the cigarette smoke are important risk factors of the inflammatory bowel disease and osteoporosis. Additionally, cigarette smoking may decrease BMD in the IBD patients. Interestingly, it affects patients with Crohn’s disease and ulcerative colitis in different ways—on the one hand it protects against ulcerative colitis, whereas on the other it increases the risk of Crohn’s disease development. Nevertheless, all patients should be encouraged to cease smoking in order to decrease the risk of developing other disorders.

## 1. Introduction

The first data regarding the noxious effects of smoking cigarettes appeared in the mid-20th century. Initially, the findings indicated that the impact of tobacco use increases the risk of lung cancer development [1]. It has been demonstrated that cigarette smoke contains various chemical compounds, including alkaloids, polycyclic aromatic hydrocarbons, and aldehydes [2,3]. In fact, the epidemiological data show that cigarette smoking increases the risk of neoplasia and cardiovascular and respiratory system diseases, and contributes to the death of about 6 million people every year. Active and passive smoking enhances the risk of diabetes mellitus type 2 [4]. Furthermore, tobacco use is associated with the risk of chronic kidney disease [5], as well as influences negatively fertility and is harmful to the fetus [6]. Cigarette smoking causes DNA methylation, which may persist even after smoking cessation [7], since the telomere lengths in smokers are shorter than in people who have never smoked, or in former smokers [8]. Moreover, cigarette smoking induces oxidative stress which is the contributing factor in a number of diseases (Figure 1), such as inflammatory bowel diseases (IBD) [9].

The IBD is diagnosed more and more frequently in some countries, particularly in Western Europe and the USA. However, the association between cigarette smoking and IBD differs in various populations [10].

Furthermore, tobacco use affects the gastrointestinal tract and increases the risk of rectal and colon cancer [11]. In fact, cigarette smoking induces cell apoptosis in the gastric mucosa, as well as inhibits the renewal of the epithelial cells in the gastrointestinal tract [2]. Moreover, according to the research, gastroesophageal reflux disease, functional dyspepsia, or irritable bowel syndrome co-existed more frequently in the smokers than in the non-smokers [12].

Electronic cigarettes (e-cigarettes) first appeared in 2003, and their use has been on the increase among young smokers [13]. However, e-cigarettes are not recommended due to the conflicting reports and insufficient epidemiological data [14]. In fact, it is possible that e-cigarettes may increase the risk of cardiovascular disease development [15].

Research suggests that over 1.1 billion people worldwide have smoked a cigarette in their lifetime. On the basis of the data from 2006, 9 million people used tobacco, and 37% and 28% of professionally active men and women, respectively, were smokers. Moreover, the frequency of cigarette smoking was different in various countries, and the rate of cigarette smoking was associated with the education level [6,16]. World Health Organization (WHO) estimates that the number of smokers in Poland will decrease to 6.6 million in 2025 [17].

The therapy of osteoporosis is based on pharmacological treatment. Drugs of choice for osteoporosis are antiresorptive agents, including bisphosphonates and denosumab. Denosumab is a fully human monoclonal antibody that inhibits osteoclastic-medicated bone resorption by binding to osteoblast-produced RANKL (Receptor Activator for Nuclear Factor κB Ligand) [18,19]. Other treatments comprise osteoanabolic drugs, for instance parathyroid hormone (PTH) derivatives (teriparatide). Additionally, nutritional support, especially an adequate intake of vitamin D and calcium, constitutes an essential element of the therapy [20]. In turn, bisphosphonates are common anti-osteoclastic drugs, but they also reduce cigarette smoking–induced osteoporotic alterations of osteoblasts and osteoclasts [21].

Moreover, OPG (osteoprotegerin) (RANKL [Receptor Activator for Nuclear Factor κB Ligand]/RANK [Receptor activator of nuclear factor-κB]) pathway is an important element of osteoporosis pathogenesis among patients suffering from IBD [22]. RANK is a protein taking part in bone metabolism [23]. RANK is produced by osteoblastic cell line (mature osteoblasts and their precursors) and activated lymphocytes T. Cytokines such as IL-1, IL-6, IL-11, and TNF-α, steroids, parathormone, and vitamin D regulate the expression of RANK [24]. In fact, inflammation and increased pro-inflammatory cytokines affect RANKL in IBD patients [25,26]. Compounds of cigarette smoke—mainly nicotine—elevate interferon γ, Il-1, and TNF-α and increase RANKL expression. Additionally, an association among cigarette smoking, RANKL/RANK/OPG pathway, and periodontitis was found. Smokers have a lower OPG level and higher RANKL/OPG ratio when compared with non-smokers [27,28]. There is no study referring to the impact of cigarette smoking on RANKL among IBD patients. However, the above-presented mechanism is likely responsible for osteoporosis in patients suffering from IBD. RANKL is expressed in osteoblasts, osteoclasts, chondrocytes, activated lymphocytes T, endothelium cells, and primary mesenchymal cells, surrounding the cartilage and chondrocytes [29]. Additionally, RANKL participates in osteoclasts differentiation and stimulation by inhibiting their apoptosis, causing increased bone resorption. RANK, when combined with RANKL, causes osteoclastogenesis from stem cells and activation of mature osteoclasts. Osteoprotegerin (OPG, TNFSF11B), produced by osteoblasts, lymphocytes, and dendritic cells, inhibits this process. Moreover, disorders of vitamin D metabolism, which occur in smoker IBD patients, may affect OPG/RANKL/RANK pathway. Patients are in a group with a higher risk of deficiency of vitamin D due to diseases. Additionally, nicotine disturbs the hydroxylation of vitamin D, affecting OPG and RANK [30]. Presented mechanisms ought to influence osteoporosis treatment among smoker patients with IBD. Supplementation of vitamin D is obligatory. Additionally, future study should be focused on treatment with the active form of vitamin D.

According to Zhu et al., bisphosphonates may be drugs for osteoporotic patients, especially smokers [21]. Many studies showed that bisphosphonates increase OPG expression and inhibit the expression of RANKL, which is opposed to the impact of nicotine on OPG/RANKL/RANK pathway [31,32]. Additionally, ECCO recommends osteoporosis treatment with bisphosphonates among IBD patients. However, there is a lack of detailed guidelines for smokers.

It is vital to notice that natural antioxidants may be useful for osteoporosis treatment. Cigarette smoking causes chronic oxidative stress, which affects bone. Therefore, foods containing antioxidants may be part of osteoporosis treatment. Vegetable, fruits, nuts, whole grain, chocolate, tea, coffee, and wine are good sources of polyphenols [33]. But patients suffering from IBD often avoid some of these products because of gastrointestinal symptoms.

Moreover, green tea polyphenols may affect BMD positively by reducing oxidative stress and inflammation. On the other hand, a high consumption of caffeine may disturb the calcium–phosphate balance. Therefore, patients at risk of osteoporosis should consume a limited portion of tea [34,35].

Noteworthy products are berries, especially maqui berries, containing bioactive compounds, among other delphinidins, which are antioxidants [36,37]. According to Zhu et al., in vitro physiological concentration of maqui extract reduced oxidative stress caused by cigarette smoking in osteoblasts. On the other hand, a high concentration of the extract was toxic for human osteoblast [37]. The animal study showed that the maqui extract inhibited body weight loss and colon shortening and decreased macroscopic and microscopic damage signs [38].

Moreover, a well-balanced diet is also an important element of osteoporosis therapy. The most discussed nutrients are vitamin D and calcium. However, sufficient intake of other diet compounds, e.g., protein, magnesium, copper, and zinc, are also essential elements of osteoporosis prevention and treatment [39,40,41].

## 2. Pathophysiological Mechanisms of Smoking on Bone Tissue

The study demonstrated that cigarette smoking decreased bone mass and increased the fracture risk [42]. Additionally, the impact of cigarette smoking on the risk of developing osteoporosis among the postmenopausal women was observed as early as in 1972, which was explained by hormonal disorders and a premature menopause [43].

Tobacco smoke affects bone mineral density both directly and indirectly [44]. Smoke compounds affect osteoclastogenesis, bone angiogenesis, body mass, calcium–phosphate balance, and adrenal and sex hormones, as well as increase oxidative stress in bone tissue, thus affecting BMD. Although osteoclasts and osteoblasts continually rebuild bone tissue, their work is regulated by a number of factors, such as hormones, cytokines, and RANK-RANKL-OPG (receptor activator of nuclear factor κβ/receptor activator of nuclear factor κβ ligand/osteoprotegerin) pathway [45]. A proper regeneration of bone tissue is conditioned by the dynamic balance in the cytokines derived from the osteoblasts: RANKL stimulates osteoclasts maturation and activity as well as OPG, which inhibits this process through binding with RANKL [46]. Tang and Lappin reported that smokers presented a lower level of OPG and a higher RANKL/OPG ratio than the non-smokers [27,28]. It was shown that nicotine might bind with nicotinic receptors of the osteoblast, thus leading to cell death [47]. Additionally, nicotine, which is a component of smoke, decreases appetite. Therefore, smokers have a lower body mass than the non-smokers [48] and a low body mass decreases the effect of the mechanical load, which is essential for the stimulation of osteogenesis. Moreover, a very lesser amount of the adipose tissue prevents extra-ovarian conversion of androgens to estrogens, and is associated with lower leptin level [49]. Nevertheless, the influence of nicotine on the differentiation of mesenchymal stem cells (MSC) to bone cell line has not been well established yet—the studies indicate both a negative and positive influence of nicotine on the proliferation and differentiation of MSC [50,51].

Theiss et al. demonstrated that the expression of collagen type I and II genes, bone morphogenetic protein (BMP), BMP-2, BMP-4, BMP-6, basal fibroblast growth factor (FGF), and the vascular endothelial growth factor (VEGF) was decreased by a systemic nicotine administration in rabbits through a posterolateral spine fusion with the autogenous bone graft [52].

Calcium–phosphate balance is essential for the bone matrix mineralization. Tobacco use influences BMD by means of affecting vitamin D and calcium absorption [53,54]. The level of serum vitamin D was significantly lower in smokers when compared to non-smokers. It is possibly caused by a lower vitamin intake and the inhibition of 25OHD hydroxylation to active form—1,25OHD—by nicotine [49]. Additionally, compounds of smoke may inhibit intestinal calcium absorption [55,56], whereas nicotine may increase serum cortisol levels [57]. In fact, cortisol affects bone cell activity and inhibits calcium absorption in the gastrointestinal tract and renal Ca reabsorption, leading to a decreased bone mass [58]. In the cigarette smoking women, smoke compounds (mainly nicotine, cotinine, and anabasine) affect enzymes, such as aromatase and hepatic 2α-hydroxylase. These enzymes are involved in the synthesis and metabolism of estrogen, which leads to a decrease in the level of active estrogen metabolites [59]. Estrogens, in turn, exhibit anabolic effects on the bone tissue [60].

The cigarette smoke exposure has been associated with a high level of free radicals, stimulating bone resorption [61]. In fact, nicotine and cotinine inhibit catalase and glutathione reductase, which causes the accumulation of reactive oxygen species. According to the research, the level of antioxidant enzymes (glutathione peroxidase, superoxide dismutase, paraoxonase) was lower in the smokers compared to the non-smokers [62]. Lee et al. reported that the activity of the nuclear factor erythroid-2-related factor-2 (Nrf2) pathway through H2O2 inhibiting a differentiation of MCT3T3 to the bone cells [63].

In contrast, the removal of Nrf2 in the bone tissue leads to a decreased mineral density due to an increased osteoclast activity and a lack of functional osteoblasts [64,65]. Nevertheless, molecular mechanisms of nicotine affecting osteoblasts apoptosis are not well known. Marinucci et al. demonstrated that nicotine stimulated intracellular hydrogen peroxide (H2O2) accumulation, thus leading to the inhibition of glyoxalase 1 (Glo1), which is an enzyme involved in the detoxification of methylglyoxal (MG). MG constitutes the primary precursor of the advanced glycation end products (AGE), which function as proapoptotic factors. Hydroimidazolone (MG-H1) constitutes an AGE, formed from the addition of the arginine to MG. The inhibition of Glo1 leads to the accumulation of MG-H1, causing over-production of H2O2 via the AGE receptor (RAGE) and a parallel apoptotic mitochondrial pathway via the induction of downregulation-dependent transglutaminase 2 (TG2). TG2, in turn, is dependent on the downregulated desensitization of nuclear factor kappa-light-chain-enhancer of the activated B cells (NF-kB). Hence, osteoporosis in the smokers may be caused by the osteoblast apoptosis resulting from the reactive oxygen species [51]. The epidemiological data, including the prevalence of IBD, osteoporosis, and smoking, are presented in Table 1.

Besides, cigarette smoking increases the Il-6 level [66], which is an important factor of osteoporosis by elevating bone resorption [67,68]. Tobacco smoking also affects adaptive immune cells, including helper T cells and CD4+CD25+ regulatory T cells [69]. Moreover, lymphocyte subsets CD4/CD8 were higher among osteoporosis patients when compared to the control group [70].

It is vital to notice that the impact of cigarette smoking on IBD development still remains unclear. Interestingly, one of the studies suggested the occurrence of genetic factors, which affect the impact of smoking on the risk of IBD [71].

## 3. Cigarette Smoking and Osteoporosis

It is widely accepted that tobacco smoke contains more than 7000 chemical compounds and that cigarette smoking contributes to an early death, tumors, and many chronic diseases. Simultaneously, smoking constitutes one of the most important modifiable factors in the development of osteoporosis, which is a growing problem in numerous countries. The mechanism is complex and includes such factors as hormonal changes and increasing oxidative stress, which further decrease bone mass and affect osteogenesis as well as angiogenesis [3]. According to the data, the smokers presented a lower serum ionic calcium and parathormone level and higher serum phosphates level than the non-smoking subjects. The concentration of 25(OH)D, alkaline phosphatase, as well as the excretion calcium to creatinine ratio were not different between the groups [104].

There were significant differences in the frequency of smoking between postmenopausal women with osteoporosis and those presenting normal BMD [60]. BMD of the forearm in the non-smoking men was higher than in the smokers [105], and the incidence of osteoporosis was also different in the smoking and non-smoking women [106]. Moreover, tobacco smoking was associated with hip fractures in women, although the risk started to decrease after ten years since quitting smoking [107]. Another study revealed that the risk of the vertebral fractures, but no other fractures, was reported, which subsequently decreased following cigarette cessation [108]. Cigarette smoking was also associated with a greater number of osteoporosis risk factors (e.g., low physical activity, low BMI, fractures in adults, and alcohol consumption) [109]. Tobacco use decreased calcium absorption and accelerated a decrease in the total and femoral neck bone mineral density [55]. Furthermore, subjects with osteoporosis smoked cigarettes more frequently than the control group (88.5% vs. 40%) [110]. Therefore, the bone mineral density T-score increased by 0.064 units for each ten years with no smoking in people who previously used tobacco [111]. In fact, a meta-analysis showed that cigarette smoking in men, both in the current and past smokers, increased the risk of hip fractures [112].

## 4. Pathophysiological Mechanisms of Smoking on Colon and Small Intestine

Cigarette smoke may affect intestines by means of various mechanisms, e.g., mucosal damage, immune response impairment, as well as intestinal microbiota disorders [113]. In addition, chronic cigarette use interferes with the protective mucus production and inhibits the repair processes in the intestinal mucus [2]. In fact, nicotine causes changes in the mucus composition in the gastrointestinal tract. This change is the consequence of increased expression of Muc2 and Muc3 in the ileum and Muc4 in the colon [114]. Moreover, cigarette smoking decreases mucosal blood flow, which may increase the inflammation in the intestinal mucosa. Simultaneously, nitric oxide (NO), included in cigarette smoke, inhibits angiogenesis leading to an impaired healing of ulcers, whereas microcirculation disorders further impair the vascular endothelial growth factor (VEGF) pathway and promote intestinal ischemia [113,115]. Thus, cigarette smoke compounds affect the integrity of the entire gastrointestinal mucosa, and the exposure to smoke results in an increased intestinal barrier permeability and in loosening of tight junctions between enterocytes [116].

Nicotine induces the production of the inflammatory chemokines and cytokines (CCR6—Chemokine receptor 6, CCL20—Chemokine C-C motif ligand 20, IL-8-Interleukin 8) in the ileum. Additionally, it changes the phenotype of the dendritic cells, including an elevated expression of MHC-II (a major histocompatibility complex class II molecules) and costimulatory molecules. Moreover, it increases the recruitment of CD4+, T CD8+, and the dendritic cells CD11b+ in the ileum [113,117].

In the IBD patients, Th1 and Th2 imbalances occur. In CD, CD4+ cells invade and produce a large number of inflammatory cytokines associated with Th1/Th17, such as IL-6, IFN-γ, IL-17A, TNF-α, and IL-23. In UC, on the other hand, CD4+ produces inflammatory cytokines related to Th2, such as IL-4, IL-13, and IL-14 [118,119,120]. In fact, nicotine changes the function of the dendritic cells, thus causing Th1 polarization in the CD patients and increasing the frequency of T Foxp3 + CD4 cells occurrence in the UC patients. Moreover, nicotine regulates the expression of T-bet (Th1 transcription factor) through α7 nicotinic acetylcholine receptor (α7nAChR) in the human lamina propria of T lymphocytes, which regulates the immune balance toward Th1 [121,122]. Moreover, cigarette smoking influences the molecular regulation expression of the cytokine mRNA and Th1 response (CCL9, CCL20, and IL-1), which are involved in the CD pathogenesis [123]. Among the UC patients, nicotine’s potentially beneficial impact has been associated with the activation of α7-nAChR in immune cells, e.g., in the macrophages and dendritic cells. A stimulation of these receptors decreases TNF-α and IL-2, which are the pro-inflammatory cytokines, and inhibits the function of CD4+ and CD25+ regulating T lymphocytes [124].

Cigarette smoking has been shown to alter an intestinal microbiota significantly, promoting opportunistic pathogens (such as some species of *Firmicutes* and *Actinobacteria)* and decreasing *Bacteroidetes* and *Proteobacteria* [125]. It is possible that the disruption of the interaction between the microbiome and the intestinal mucosa constitutes one of the major etiological factors of IBD. Therefore, cigarette smoking may influence the occurrence of IBD.

Nicotine and other cigarette smoke compounds are the important risk factors of the inflammatory bowel disease and osteoporosis. In fact, cigarette smoking may decrease BMD in the IBD patients. As it has already been mentioned, nicotine induces immune mechanisms in the intestine leading to an increase of the inflammation, which affects the RANK-RANKL-OPG pathways, thus stimulating osteoclastogenesis in patients with IBD [126]. Additionally, an increased inflammation in the intestinal mucosa leads to an exacerbation of the disease. It has been associated with the treatment intensification, for instance, the use of steroids which constitute risk factors of osteoporosis. Furthermore, the compounds in the cigarette smoke may increase the activity of the disease and cause hormonal disorders, mainly a decrease of BMD-protecting estrogen levels [127]. In the smoking IBD patients, the intestinal mucosa damage leads to a calcium absorption disorder and a loss of nutrients, such as proteins, which causes body weight loss and malnutrition, resulting in a decreased bone mass. Therefore, patients with IBD are a group particularly vulnerable in terms of the negative effects of cigarette smoking on bones. Therefore, it is suggested that among smokers suffering from IBD, bone may be doubly affected negatively—firstly, due to the disease, and secondly because of the cigarette smoke impact.

## 5. Cigarette Smoking and IBD

The smoking behavior among the IBD patients is similar to healthy individuals. Nevertheless, the smokers were treated biologically or surgically more frequently than the non-smokers [128]. Moreover, there are 64 single nucleotide polymorphisms associated with IBD, which are modified by the tobacco use [71]. In fact, a meta-analysis demonstrated an association between cigarette smoking in the past and the risk of ulcerative colitis development. Interestingly, a current tobacco use may protect from the development of UC. However, both the current and past cigarette smoking was linked with Crohn’s disease [129]. Moreover, tobacco use additionally increased the risk of microscopic colitis [130], as well as decreased ulcer healing. In mice, smoke exposure increased the level of dendritic cells, macrophages, CD4+, and CD8+ T cells [113]. Nevertheless, no association between passive smoking in childhood and the risk of development of UC or CD was observed [131]. Furthermore, cigarette use increased the risk of surgery, re-operation, and medical therapy (e.g., steroids) in Crohn’s disease [132]. In fact, the CD patients who stopped smoking relapsed less often than the patients who continued to use tobacco [133]. It is vital to bear in mind that according to the studies, IBD constituted a risk factor of osteoporosis and, additionally, cigarette smoking increased the risk of a low BMD in patients with CD and UC [134]. Moreover, chronic obstructive pulmonary disease increased the risk of inflammatory bowel diseases [135]. In CD, the risk of restrictive procedures of the intestinal tract was higher in the smokers than non-smokers, although the study revealed no differences between the former and current smokers. No higher risk of colectomy was observed in the smokers than non-smokers; nevertheless, former smokers had an increased risk of colectomy compared to individuals who never smoked [136]. In CD, the smokers presented more complications than never smokers [137]. The study showed 19 single nucleotide polymorphisms (SNPs) associated with Crohn’s disease, 25 SNPs linked to UC, and 25 SNPs connected to IBD were affected by cigarette smoking [71]. In the smoking Crohn’s disease patients, the course of the disease was more severe, and immunosuppressive treatment was more frequently needed than in the non-smokers. The age of CD diagnosis was lower in the smokers than in the former smokers and the non-smokers (not statistically significant). In UC, the diagnosis age was higher in the ex-smokers and the non-smokers as compared with the smokers [138]. Additionally, in the IBD patients, adolescents knew about the harmful effects of cigarette smoking, and young individuals suffering from CD knew more about the influence of tobacco use on the course of the disease than UC patients [103]. Interestingly, nicotine—or its metabolites—may affect healing in the UC patients. Furthermore, using a chewing gum with nicotine may make the cessation of smoking easier for the patients without influencing the disease course [113].

## 6. Conclusions

In conclusion, on the basis of the review, we think that cigarette smoking is a modifiable risk factor of osteoporosis. Seeking methods of treating osteoporosis that are safe for patients is needed. Additionally, research into natural products in the treatment of osteoporosis is promising. Nevertheless, patients should be encouraged to smoking cessation. Therefore, recommendations for the IBD patients should include information regarding the influence of cigarette smoking on the risk of osteoporosis together with a recommendation on smoking cessation:Cigarette smoking is a risk factor of the development of and a deterioration in the course of Crohn’s disease. Tobacco use protects from the development of ulcerative colitis. However, due to other smoking-related conditions, cigarette smoking is not recommended both in CD and UC [139].Guidelines of AACE/ACE (American Association Of Clinical Endocrinologists And American College Of Endocrinology) report clearly that cigarette smoking constitutes a risk factor of osteoporosis development, and all patients should be persuaded to quit smoking [140].European guidelines also contain data concerning cigarette smoking as a risk of osteoporosis [141].

## Figures and Tables

**Figure 1 jcm-10-01515-f001:**
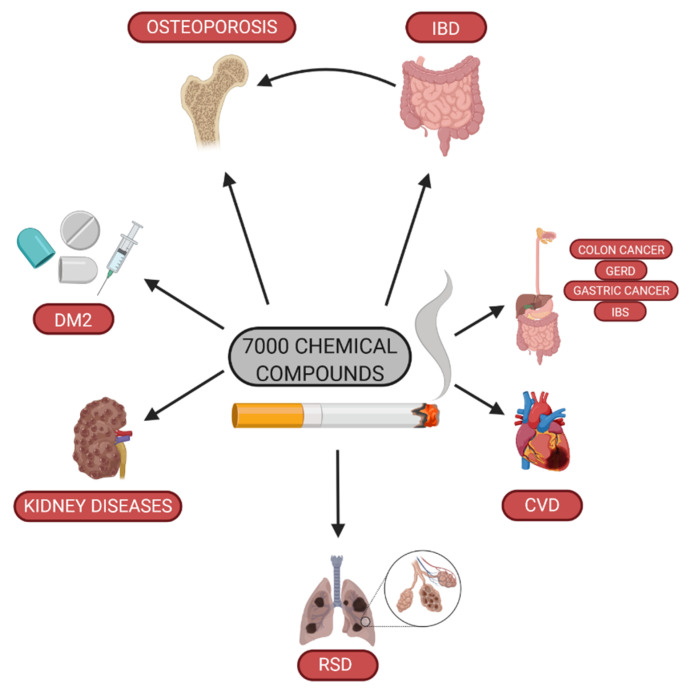
Impact of cigarette smoking on health (DM2—diabetes mellitus type 2; RSD—respiratory system disease; CVD—cardiovascular disease; GERD—gastroesophageal reflux disease; IBS—irritable bowel syndrome; IBD—inflammatory bowel diseases).

**Table 1 jcm-10-01515-t001:** Epidemiological data including the prevalence of IBD, osteoporosis in IBD, smokers among the IBD patients, and morbidity of IBD in various regions and countries.

Country	Prevalence of IBD (%)	Ref.	Prevalence of Osteoporosis in IBD (%)	Ref.	Percentage of IBD Smokers (%)	Ref.	Morbidity (*n*/100000)	Ref.
Europe	0.006(CD) 0.010 (UC)	[72]	no data		24.0–73.0 (CD) 5.4–64.0 (UC)	[73]	12.7 (CD) 24.3 (UC)	[72]
North America	0.420 (IBD) 0.040–0.250 (UC)	[74]	no data		4.0 (CD) 8.6 (UC)	[73]	0–20.2 (CD) 0–19.2 (UC)	[75]
Australia	0.030 (for CD) 0.020 (for UC)	[76]	no data		15.0 (CD) 5.0 (UC)	[77]	29.3 (CD) 17.4 (UC)	[78]
Finland	0.120 (for CD) 0.290 (for UC)	[79]	no data		no data		9.2 (CD) 24.8 (UC)	[80]
Hungary	0.050 (for CD) 0.140 (for UC)	[81]	no data		no data		4.7 (CD) 11.0 (UC)	[82]
United Kingdom	0.280 (for CD) 0.430 (for UC)	[83]	11.6–13.6 (CD)	[84]	32.2 (CD)	[84]	10.2 (CD) 15.7 (UC)	[85]
Germany	0.320 (for CD) 0.410 (for UC)	[86]	15.0 (CD) 7.0 (UC)	[87]	16.1 (CD) 7.5 (UC) upon diagnosis	[88]	6.1 (CD) 3.9 (UC)	[88]
Slovakia	0.680 (CD)	[89]	10.0 (UC) 15.2 (CD)	[90]	42.0 (CD) 36.0 (UC) upon diagnosis	[91]	no data	
Netherlands	0.230 (CD) 0.280 (UC)	[92]	28.0 (CD) 11.0 (UC)	[93]	46.1 (CD) 20.0 (UC)	[94]	17.2 (CD) 10.5 (UC)	[78]
Spain	0.190 (CD) 0.350 (UC)	[95]	17.0 (CD) 27.7 (UC)	[96]	66.7 (CD) 12.5 (UC)	[97]	10.8 (CD) 9.4 (UC)	[98]
Canada	0.370 (CD) 0.300 (UC)	[99]	7.0 (IBD)	[100]	no data		23.8 (CD) 23.1 (UC)	[78]
United States	0.900 (IBD)	[101]	no data		no data		6.3 (CD) 8.8 (UC)	[78]
Poland	0.160 (IBD) 0.040 (CD)	[75]	5.8–11.7 (CD) 2.9–3.8 (UC)	[102]	12.2 (IBD) 8.3 (CD) 18.2 (UC) /children and adolescents/	[103]	no data	

## Data Availability

Not applicable.

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
