# Peer review of "Impact of Cigarette Smoking on the Risk of Osteoporosis in Inflammatory Bowel Diseases"

_jcm, 2021, doi:10.3390/jcm10071515_

Round 1

Reviewer 1 Report

The topic of the present study is to illustrate the effect of cigarette smoking on bone among IBD patients, which is a novel and interesting point. However, the topic is not well expressed in the text. Detailed comments are as follow:
1. The title of this review is to focus on the effect of cigarette smoking on bone in IBD patients, which is a relatively new topic. However, most of the text is devoted to the harmful effect of cigarette smoking, the relationship between smoking and osteoporosis, and the relationship between IBD and osteoporosis. These aspects are literally known and have been well demonstrated previously. This review should address the specific effects of smoking on the skeletal system of patients with IBD, and the above aspects can be briefly mentioned in the introduction. The effect of cigarette smoking on bone changes in IBD patients was missing.
2. As mentioned, there is not innovative to emphasize the relationship between cigarette smoking and osteoporosis and the relationship between IBD and osteoporosis. What is interesting are: Are there differences in bone changes between IBD patients who smoke and those who do not smoke? What are the reasons for this? What are the potential treatment options for IBD patients who smoke?
3. “Key words” are not really appropriate. “Crohn’s disease”, “Ulcerative colitis” and “IBD in COVID-19 era” were not the major aspects in this review. In contrast, “Osteoporosis” was missing. Specifically, “IBD in COVID-19 era” was not found in the text, only one paragraph introduced the relationship between cigarette smoking and COVID-19. In fact, this paragraph also seems redundant in this review.
4. The authors introduced the mechanisms of cigarette smoking on the bones and IBD respectively. However, there is no comprehensive analysis of these mechanisms. Which ones are important in IBD smokers? Is there any treatment that targets one of these mechanisms to treat IBD smokers? What are the latest research advances?
5. There is too little personal perspective from the authors, only facts from the literature can be obtained in each section.
6. For conclusions such as smoking is a risk factor for osteoporosis and smoking cessation is recommended for patients with IBD and osteoporosis, these are not new and significant. Besides, the conclusion seems not to be related to the title of this review.

In conclusion, the topic of this review is relatively attractive and innovative, but the content is not focused on the topic and too much context is devoted to content that is widely known and not relevant to the topic. According to the title of this review, the authors should be aware of these questions: Why do we need to be concerned about the effects of cigarette smoking on the bones of patients with IBD? What are the differences between smoking in non-IBD patients and IBD patients? How do the effects of IBD on bone differ between the smokers and non-smokers? What are the underlying mechanisms? What treatment options or potential treatments are available to prevent bone disease in IBD patients who smoke?

Reviewer 2 Report

This is a relevant and well-presented review on the role of cigarette smoking on bone mineral density in IBD patients. The overall structure of the review is adequate and from what I can judge, the authors correctly cite the relevant literature and bring it into perspective.

As a minor comment: Figure 1 appears twice. Further, the second figure does not add or summarize any substantial information and may be omitted entirely. 

The manuscript requires extensive editing of English language and style.

Round 2

Reviewer 1 Report

Dear authors,

Thank you for your responses.

I appreciate you responding to my comments point by point, however, valuable changes in the manuscript were not seen except for language editing.

Let me reiterate a few important points:

  1. First and foremost, it was nothing new that cigarette smoking is an important risk factor for osteoporosis and IBD and to quit smoking can prevent osteoporosis. Therefore, if the conclusion of this review only reveals these points would be less significant.

You have addressed some interesting aspects about IBD and osteoporosis in other reviews like nutrition (PMID: 33562891; PMID: 33498571; PMID: 33451170), something new or at least controversial should be illustrated in this review.

  1. The questions I raised in the first-round comments is a clue for you to revise the manuscript. It is true that not many studies focused on your topic, but if you want to stick to this topic you have to really dig out interesting and significant points.

Actually, something interesting still could be discussed in your review.

As you mentioned in your response letter, Inflammation and proinflammatory interleukins, malabsorption and deficiency of calcium and vitamin D could be a link between cigarette smoking, IBD, and osteoporosis. Please add these potential mechanisms into manuscript.

Talking about treatments, which anti-osteoporosis drugs (bisphosphonates, SERMS, teriparatide, denosumab etc.) are suitable for smoker of IBD can be discussed.

Actually, some novel and potential treatments like natural anti-oxidant (e.g. Maqui berry, green tea), bisphosphonates and nutrition treatments are recently demonstrated to be beneficial for cigarette-affected osteoporosis. Are these treatments suitable for IBD patients as well?

The author well-summarized the underlying mechanisms between cigarette and osteoporosis and between cigarette and IBD. If the two parts can be linked to find some common pathways, this will be of great significance for future research revelations.

In a word, something new or perspective ideas need to be added. These aspects may not be conclusive, but it needs to be integrated and summarized by the author in an organized way. These points will be more attractive than the known knowlegements.
